# Prompt-Guided Image-Adaptive Neural Implicit Lookup Tables for Interpretable Image Enhancement

## ABSTRACT

In this paper, we delve into the concept of interpretable image enhancement, a technique that enhances image quality by adjusting filter parameters with easily understandable names such as "Exposure" and "Contrast". Unlike using predefined image editing filters, our framework utilizes learnable filters that acquire interpretable names through training. Our contribution is two-fold. Firstly, we introduce a novel filter architecture called an image-adaptive neural implicit lookup table, which uses a multilayer perceptron to implicitly define the transformation from input feature space to output color space. By incorporating image-adaptive parameters directly into the input features, we achieve highly expressive filters. Secondly, we introduce a prompt guidance loss to assign interpretable names to each filter. We evaluate visual impressions of enhancement results, such as exposure and contrast, using a vision and language model along with guiding prompts. We define a constraint to ensure that each filter affects only the targeted visual impression without influencing other attributes, which allows us to obtain the desired filter effects. Experimental results show that our method outperforms existing predefined filter-based methods, thanks to the filters optimized to predict target results. We will make our code publicly available upon acceptance.

## CCS CONCEPTS

• **Computing methodologies** → **Image processing**; **Computational photography**.

## KEYWORDS

Image enhancement, Lookup table, Implicit neural representation, Vision and language, CLIP, Interpretablity

## 1 INTRODUCTION

Image enhancement has become an essential task in modern digital image processing, enhancing the visual quality of images by adjusting their brightness and color. This process significantly increases an image's utility across various applications. This paper focuses on image enhancement techniques, examining their scope and potential in detail. Especially, we delve into the concept of interpretable image enhancement, a technique that improves images through the adjustment of filter parameters with easily understandable names, such as "Exposure", "Contrast", and "Saturation". This

Permission to make digital or hard copies of all or part of this work for personal or classroom use is granted without fee provided that copies are not made or distributed for profit or commercial advantage and that copies bear this notice and the full citation on the first page. Copyrights for components of this work owned by others than the author(s) must be honored. Abstracting with credit is permitted. To copy otherwise, or republish, to post on servers or to redistribute to lists, requires prior specific permission and/or a fee. Request permissions from permissions@acm.org.
*ACM MM, 2024, Melbourne, Australia*
© 2024 Copyright held by the owner/author(s). Publication rights licensed to ACM.
ACM ISBN 978-x-xxxx-xxxx-x/YY/MM
https://doi.org/10.1145/nnnnnnn.nnnnnnn

approach allows the user to adjust the enhancement results according to his or her preference and to learn and more effectively utilize the image enhancement process itself. Consequently, interpretable image enhancement is anticipated to substantially enhance users' comprehension and manipulation of image processing.

Previous interpretable image enhancement methods [11, 15, 23, 24] employ predefined image editing filters, and convolutional neural networks (CNNs) are trained to determine the optimal parameters for these filters. Since these filters are designed in a manner that is understandable to humans, they facilitate interpretable image enhancement. However, the effectiveness of enhancement may be constrained by the limitations inherent in the design of these predefined filters. For instance, the "Exposure" filter can be designed in various ways, making it challenging to manually craft an optimal Exposure filter for achieving specific results. In contrast, most recent image enhancement methods [21, 35, 36, 39, 40] employ 3D lookup tables (LUTs) [37], which are tables that record input RGB values and corresponding output RGB values. Multiple 3D LUTs are employed to apply various effects, and image-adaptive enhancement is achieved by linearly summing these 3D LUTs, weighted by image-adaptive parameters. Unlike predefined image editing filters, 3D LUTs are learnable filters optimized for predicting enhancement results, enabling high quality enhancement. However, there are two notable issues associated with the use of 3D LUTs. Firstly, the expressive power is limited. This is because the multiple 3D LUTs are merely summed in a linear fashion, weighted by image-adaptive parameters, which means the image-adaptive parameters can only adjust the enhancement effect in a linear manner. Secondly, 3D LUTs lack interpretable names. Since they are optimized solely for predicting target enhancement results, their effects may not be intuitively understood by humans.

To achieve high-performing and interpretable enhancement methods, we propose learnable and interpretable filters named a Prompt-Guided Image-Adaptive Neural Implicit Lookup Table (PG-IA-NILUT). Our contribution is twofold. Firstly, we introduce a novel learnable filter architecture called an Image-Adaptive Neural Implicit Lookup Table (IA-NILUT). Inspired by a previous method [7], we utilize implicit neural representations [28] for a color transformation. While previous researchers have used 3D LUTs to explicitly record input-output RGB value pairs, we employ a multilayer perceptron (MLP) to implicitly define the transformation from input feature space to output color space. The most significant distinction from the 3D LUT-based methods is that we incorporate image-adaptive parameters directly into the input features. Since an MLP can represent nonlinear and complex relationships between inputs and outputs, our approach enables these image-adaptive parameters to exert a complex influence on the output RGB values, thereby achieving highly expressive filter effects. Additionally, to address the problem of high computational costs of MLPs, we introduce the

technique called LUT bypassing. Instead of applying the MLP directly to each pixel in the image, we convert the MLP into a 3D LUT, which is then applied to each pixel. Color transformation through the 3D LUT is computationally inexpensive, enabling cost-effective image enhancement.

As a second contribution, we propose a prompt guidance loss to assign interpretable names to each filter. This loss function utilizes CLIP [25], a vision and language model capable of embedding images and text within the same feature space. CLIP has demonstrated its ability to quantify image impressions [30]. For example, for an impression word such as "Exposure," we prepare pairs of positive and negative prompts (e.g., "Overexposed photo." and "Underexposed photo.") and calculate the ratio of the distances between the image feature and each prompt feature. This allows us to quantitatively evaluate the "Exposure" impression conveyed by the image. In this study, we propose using the pairs of positive and negative prompts as guiding prompts to guide the filters toward achieving the desired effects. Our prompt guidance loss ensures that when the parameter associated with "Exposure" is altered, only the "Exposure" score changes, while the scores for other impressions remain unaffected. By minimizing this prompt guidance loss in conjunction with a reconstruction loss of the target results, we achieve high-performing and interpretable filters.

To evaluate the proposed method, we perform experiments with the FiveK [4] and PPR10K [20] datasets. We show that the proposed method achieves interpretable filters, which are understandable to humans. In addition, the proposed method achieves higher performance than existing predefined filter-based methods.

The contributions of this paper are as follows:

- For interpretable and learnable filters, we develop the IA-NILUT, a highly expressive filter architecture.
- We introduce the prompt guidance loss to assign interpretable names to each filter.
- The proposed method achieves higher performance than existing predefined filter-based methods.

## 2 RELATED WORKS

### 2.1 Encoder-Decoder-Based Methods

Early CNN-based image enhancement methods utilized encoder-decoder-based CNNs. Gharbi et al. [8] achieved rapid image enhancement using a bilateral grid [5]. Chen et al. [6] modified U-Net [26] to incorporate global features. Wang et al. [31] introduced a loss function for spatial smoothness. Moran et al. [22] proposed a lightweight model that employs local parametric filters. Kim et al. [13] developed a sequential approach to image enhancement, applying global and local adjustments in stages. Afifi et al. [1] presented a versatile model capable of correcting both overexposed and underexposed images. Kim et al. [12] developed a representative color transform technique for improved color accuracy. Zhao et al. [42] explored the use of invertible neural networks to restore content accurately while avoiding bias. Zhang et al. [41] leveraged Transformer [29] for structure-aware enhancement. Recognizing the diversity in user preferences, some researchers have focused on personalized image enhancement models [14, 17]. Because encoder-decoder-based methods are computationally costly, filter-based approaches have recently become more prevalent.

### 2.2 Predefined Filter-Based Methods

Predefined filter-based methods train CNNs to predict the parameters of predefined image editing filters. Park et al. [24] employed reinforcement learning to train an agent that iteratively determines the parameters. Hu et al. [11] utilized generative adversarial networks (GANs) to generate more realistic results. Kosugi and Yamasaki [15] reproduced Photoshop filters, enabling more efficient prediction of enhancement results. Bianco et al. [3] and Li et al. [19] used color transformation curve for flexible enhancement. Ouyang et al. [23] achieved local enhancement with region-specific color filters. Some researchers proposed methods for crowd workers to adjust the filter parameters [16, 18]. These methods can achieve interpretable enhancements because the predefined filters are named in a way that is understandable to humans, but the enhancement performance can be limited by the design of these predefined filters.

### 2.3 Learnable Filter-Based Methods

Learnable filter-based methods optimize the filters using training data. He et al. [9] successfully replicated an image editing process using an MLP. Wang et al. [33] further enhanced these results by applying sequential image retouching.

Recent learnable filter-based methods largely use 3D LUTs, which are trainable tables that map input RGB values to corresponding output values. Zeng et al. [37] utilized multiple 3D LUTs, combining them with image-adaptive weights. Wang et al. [32] introduced spatial-aware 3D LUTs. Yang et al. [35] made the sampling points of 3D LUTs adapt to images, enhancing their expressiveness. Yang et al. [36] incorporated a 1D LUT alongside 3D LUTs for complex color transformations. Zhang et al. [39] proposed a compressed representation of 3D LUTs to efficiently increase their number. Zhang et al. [40] introduced hashing techniques to reduce parameters. Liu et al. [21] defined 4D LUTs for local enhancement. Shi et al. [27] developed a network that considers cross attention between RGB values and LUTs. Zhang et al. [38] combined 3D LUTs with local laplacian filters [2] for advanced effects. Despite the high performance, they lack interpretability, presenting a challenge for understanding the modifications they make to the images.

## 3 PRELIMINARY

This section describes the key existing method: image-adaptive 3D LUTs [37]. 3D LUTs are learnable tables that record input RGB values and the corresponding output RGB values. We denote the matrix representing the sampling points by $\mathbf{I} \in \mathbb{R}^{N^3 \times 3}$ and the matrix recording the corresponding output values by $\mathbf{O} \in \mathbb{R}^{N^3 \times 3}$, where $N$ is the number of sampling coordinates. Given an input RGB value of $[r^x, g^x, b^x]$, an index $s$ is searched for such that the vector in the $s$-th row of $\mathbf{I}$ matches $[r^x, g^x, b^x]$; then, the $s$-th row of $\mathbf{O}$, denoted as $[r^y, g^y, b^y]$, is returned. If the input RGB value is not included in $\mathbf{I}$, an interpolated value is returned based on the surrounding RGB values. This process is performed on all pixels. Let $\mathbf{X}$ and $\mathbf{Y}$ be input and output images, respectively, the transformation is represented as $\mathbf{Y} = \text{Lookup}(\mathbf{X}, \{\mathbf{I}, \mathbf{O}\})$.

In image-adaptive 3D LUTs [37], multiple LUTs $\{\mathbf{I}, \mathbf{O_1}\}, ..., \{\mathbf{I}, \mathbf{O}_J\}$ are employed for different effects. To achieve the optimal enhancement for each image, CNN-based weights predictor F is trained to output weights for each LUT as $\mathbf{w} = \text{F}(\mathbf{X})$, where $\mathbf{w} \in \mathbb{R}^J$. The

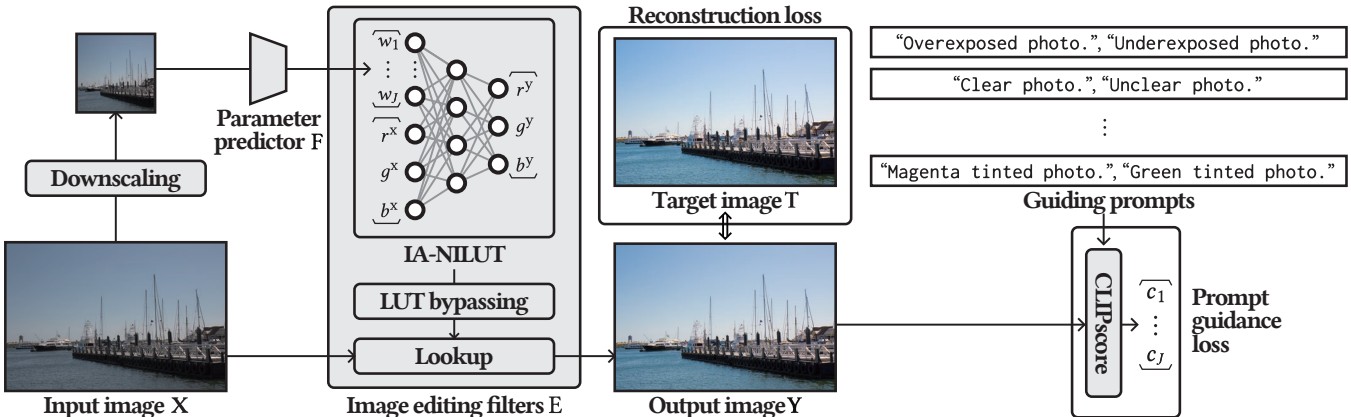

**Figure 1: Overview of our interpretable image enhancement method. For a highly expressive filter architecture, we propose an IA-NILUT. By employing LUT bypassing, we can expedite the transformation process. Additionally, we introduce a prompt guidance loss to assign interpretable names to each filter. As our method provides an interpretable and learnable framework for enhancement, it outperforms other predefined filter-based methods in terms of performance.**

enhanced result is represented as

$$Y = \text{Lookup}(X, \{I, O_1\}) \times w_1 + \cdots + \text{Lookup}(X, \{I, O_J\}) \times w_J. \quad (1)$$

Each $\text{Lookup}(X, \{I, O_j\})$ can be regarded as the result of applying different filters to $X$, and each $w_j$ works as a filter parameter that determines the strength of the filter effect. $O_1, ..., O_J$ can be optimized to predict enhancement results, which makes the lookup tables as efficient image editing filters. Because pixels are transformed independently, Eq. (1) can be simplified as

$$Y = \text{Lookup}(X, \{I, O_1 \times w_1 + \cdots + O_J \times w_J\}). \quad (2)$$

The weights predictor F processes images that are downscaled to a fixed size, and Lookup function operates quickly. As a result, this framework enables real-time enhancement for images of any size.

This approach faces two main issues. First, there's the issue of limited expressive power. The enhancement results are summed linearly as shown in Eq. (1), meaning that image-adaptive parameters cannot produce complex effects. Second, the 3D LUTs lack interpretable names. Since the 3D LUTs are optimized solely for predicting target results, there's no assurance that their effects will be meaningful or understandable to humans. We address these challenges by introducing highly expressive and interpretable filters.

## 4 PROPOSED METHOD

To achieve a high-performing and interpretable image enhancement method, we make two contributions. First, we propose a novel learnable filter architecture called an IA-NILUT. Second, we introduce a prompt guidance loss to give interpretable names to each filter. We show the overview in Figure 1 and describe the contributions in the following sections.

### 4.1 Image-Adaptive Neural Implicit Lookup Table

We propose a novel filter architecture called an IA-NILUT. Inspired by the existing method known as NILUTs [7], our approach employs

an implicit neural representation [28], wherein we implicitly define the transformation from input space to output space using an MLP. We visualize the difference between the 3D LUTs and our IA-NILUT in Figure 2. The most significant distinction between the previous image-adaptive 3D LUTs and our IA-NILUT is that the IA-NILUT incorporates the image-adaptive parameters directly into the input features. Given that an MLP is capable of capturing nonlinear and intricate relationships between input and output variables, our method allows the image-adaptive parameters to intricately affect the output RGB values, thereby achieving highly expressive filter effects. We define the color transformation process as follows,

$$\begin{aligned}
[r^y, g^y, b^y] = [r^x, g^x, b^x] \\
+ \text{e}\big([r^x, g^x, b^x] \oplus \text{sort}([r^x, g^x, b^x]) \oplus \mathbf{w}\big) \quad (3) \\
- \text{e}\big([r^x, g^x, b^x] \oplus \text{sort}([r^x, g^x, b^x]) \oplus \mathbf{0}\big),
\end{aligned}$$

where e represents the MLP, and $\oplus$ denotes vector concatenation. We make two improvements to the color transformation process for interpretable filters. First, we use the sorted RGB values, which are denoted by $\text{sort}([r^x, g^x, b^x])$, because they play an important role in filter interpretability. For instance, in the HSV color space, saturation is determined by the maximum and minimum RGB values. Second, we add the difference between $\text{e}\big([r^x, g^x, b^x] \oplus \text{sort}([r^x, g^x, b^x]) \oplus \mathbf{w}\big)$ and $\text{e}\big([r^x, g^x, b^x] \oplus \text{sort}([r^x, g^x, b^x]) \oplus \mathbf{0}\big)$ into the input RGB values. This ensures that the original RGB values are retained in the output when $\mathbf{w}$ is set to $\mathbf{0}$, a common characteristic of image editing filters. We define $\hat{\text{E}}$ as a function that applies Eq. (3) to each pixel of image $X$, and the image transformation process is represented as follows:

$$Y = \hat{\text{E}}(X, \mathbf{w}). \quad (4)$$

**LUT bypassing.** Since MLPs involve multiple nonlinear transformations, the computational cost is significant, especially when processing large sized images. To address this issue, we propose LUT bypassing. Instead of directly applying the MLP to every pixel of the image, we convert the MLP into an LUT and apply this LUT to

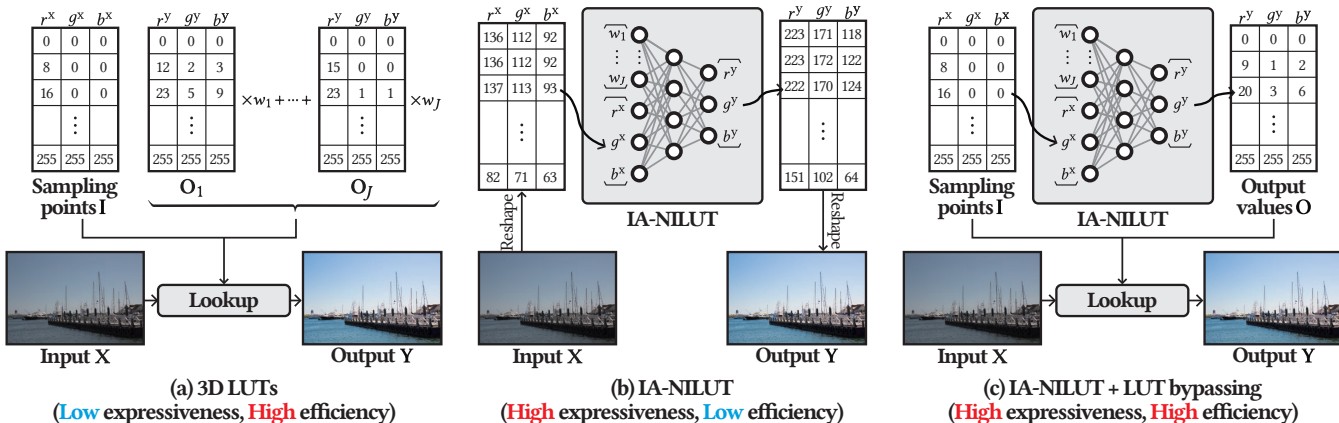

Figure 2: Comparison between the 3D LUTs [37], the IA-NILUT, and the IA-NILUT with the LUT bypassing.

the image as shown in Figure 2(c). Eq. (4) is transformed as follows,

$$Y = E(X, w) = \text{Lookup}(X, \{I, O\}),$$
$$\text{where } O = \hat{E}(I, w). \tag{5}$$

The sampling points $I \in \mathbb{R}^{N^3 \times 3}$ are considered as an image with $N^3$ pixels. This is then converted into $O$ by the MLP. Following this conversion, the input image $X$ is transformed using the lookup table comprising pairs of $I$ and $O$. In our experiment, we set $N$ to 33, which results in $I$ being treated as an image composed of 35,937 pixels. For comparison, a $512 \times 512$ image contains 262,144 pixels, indicating that $I$ represents a relatively small image. Even when processing large-sized images, the MLP is applied only to $I$, which means that the computational cost of the MLP remains constant. Therefore, LUT bypassing leverages the expressive power of MLPs while also benefiting from the low computational cost associated with LUTs.

**Comparison with advanced LUT-based methods.** Recent researchers have made various improvements to LUTs to enhance their expressiveness. For example, AdaInt [35] makes the sampling points $I$ to be image-adaptive. CLUTNet [39] uses a compressed representation of 3D LUTs. The most significant difference between our method and these existing methods lies in the number of image-adaptive parameters. The existing methods improve expressiveness by increasing the number of image-adaptive parameters; for example, AdaInt and CLUTNet use 99 and 20 image-adaptive parameters, respectively. However, this approach makes interpretability more complex. Too many parameters can make the image editing process confusing for users. In contrast, our method boosts expressiveness by using an implicit neural representation, without increasing the number of image-adaptive parameters. In our experiments, we use only five image-adaptive parameters. This results in a filter architecture that's easier to understand.

### 4.2 Prompt Guidance Loss

We introduce a prompt guidance loss that assigns interpretable names to each filter. In this loss function, we utilize CLIP [25], a vision and language model that embeds images and text within

the same feature space. CLIP has demonstrated its ability to quantitatively assess visual impressions [30]. When evaluating an image's "Exposure," we create pairs of prompts that contrast positive and negative aspects, such as "Overexposed photo." versus "Underexposed photo." We denote the distances between the image feature and each prompt feature as $d^+$ and $d^-$, respectively. The image's Exposure impression can be evaluated using the formula $\exp(d^+)/(\exp(d^+) + \exp(d^-))$.

We propose using the pairs of positive and negative prompts as guiding prompts to guide the filters toward achieving the desired effects. We illustrate our motivation in Figure 3. We prepare $J$ filter names along with pairs of corresponding guiding prompts, assigning a filter name to each dimension of the $J$-dimensional image-adaptive parameters $w$. During the training phase, we assess the impressions of the enhanced results with each guiding prompt. When we assign the filter name "Exposure" to $w_1$, we expect that a change in $w_1$ will only affect the Exposure score, without impacting other scores such as "Contrast" or "Saturation" as shown in Figure 3(b). If the Contrast and Saturation scores change as shown in Figure 3(c), this could be considered undesired behavior for the Exposure filter, potentially confusing users. Therefore, we propose a constraint that ensures only specific scores are affected when parameters are altered, while other scores remain unchanged.

We define randomly sampled weights as $\overline{w}$, and denote the scores $c \in \mathbb{R}^J$ evaluated on $J$ prompt pairs as follows.

$$c = \text{CLIPscore}\big(E(X, \overline{w})\big). \tag{6}$$

To ensure that a specific filter effect is applied when $w_j$ is altered, we define the prompt guidance loss. Instead of adding $\Delta \overline{w}_j$ to $\overline{w}_j$, we directly apply a constraint to the gradient in the following way.

$$\mathcal{L}_{\text{PG}} = \sum_{j=1}^{J} \left( \lambda_j \left| \frac{\partial c_j}{\partial \overline{w}_j} - 1 \right| + \lambda \sum_{j' \neq j} \left| \frac{\partial c_{j'}}{\partial \overline{w}_j} - 0 \right| \right), \tag{7}$$

where $\lambda_j$ and $\lambda$ are hyperparameters. This constraint guarantees that $w_j$ affects only the targeted score $c_j$, while the remaining scores $c_{j'}(j' \neq j)$ are unaffected. By minimizing $\mathcal{L}_{\text{PG}}$, we can assign interpretable names to each filter.

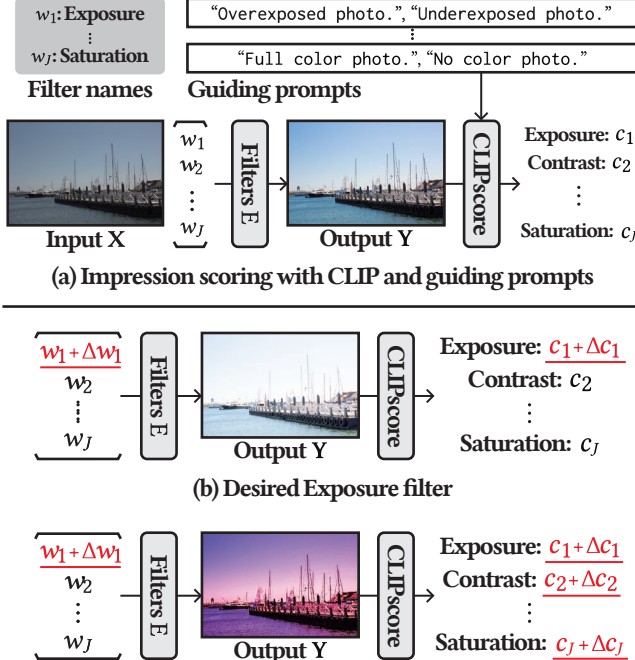

(a) Impression scoring with CLIP and guiding prompts

(b) Desired Exposure filter

(c) Undesired Exposure filter

Figure 3: Motivation for our prompt guidance loss.

## 4.3 Training and Testing

The pairs of input and target images for training are denoted as $\{X_1, T_1\}, \ldots, \{X_I, T_I\}$. We divide the training steps into three stages. In the first training stage, only the filters E are trained, using only the prompt guidance loss $\mathcal{L}_{PG}$.

$$E = \underset{E}{\operatorname{argmin}} \ \mathcal{L}_{PG}. \tag{8}$$

In the second stage, we introduce image-adaptive parameters $\mathbf{w}_1, \ldots, \mathbf{w}_I$ for the images $X_1, \ldots, X_I$. The training process is defined as

$$E, \mathbf{w}_1, \ldots, \mathbf{w}_I = \underset{E, \mathbf{w}_1, \ldots, \mathbf{w}_I}{\operatorname{argmin}} \ \mathcal{L}_{PG}$$
$$+ \sum_{i=1}^{I} \mathrm{MSE}(\mathbf{T}_i, E(\mathbf{X}_i, \mathbf{w}_i)) + \sum_{i=1}^{I} \mathrm{MSE}(\mathbf{X}_i, E(\mathbf{T}_i, -\mathbf{w}_i)), \tag{9}$$

where MSE represents the mean squared error function. The third term is a constraint ensuring that the input image is reconstructed from the target image when the parameters $\mathbf{w}_i$ are reversed, a property that existing filters also possess. In the final stage, the parameter prediction model F is trained as

$$F = \underset{F}{\operatorname{argmin}} \ \sum_{i=1}^{I} \mathrm{MSE}(\mathbf{T}_i, E(\mathbf{X}_i, F(\mathbf{X}_i))). \tag{10}$$

At test time, the enhancement results are generated using the trained E and F, as $\mathbf{Y} = E(\mathbf{X}, F(\mathbf{X}))$. The filters E can achieve fast transformations through the LUT bypassing, and the parameter prediction model F resizes the input image to a fixed resolution before processing, resulting in real-time enhancement.

## Table 1: Guiding prompts.

| | Filter name | Positive prompt | Negative prompt |
|---|---|---|---|
| $w_1$ | Exposure | "Overexposed photo." | "Underexposed photo." |
| $w_2$ | (FiveK) Contrast (PPR10K) Contrast | "Clear photo." "High contrast photo." | "Unclear photo." "Low contrast photo." |
| $w_3$ | Saturation | "Full color photo." | "No color photo." |
| $w_4$ | Color temperature | "Yellow tinted photo." | "Blue tinted photo." |
| $w_5$ | Tint correction | "Magenta tinted photo." | "Green tinted photo." |

## 5 EXPERIMENTS

### 5.1 Datasets and Implementation

We utilize two widely used datasets: FiveK [4] and PPR10K [20]. FiveK contains 5,000 images, each retouched by five experts. Following the setting of previous papers [35, 39], we use 4,500 of these images for training and the remaining 500 for testing, employing the images retouched by Expert C as the target images. We conduct experiments in both 480p resolution (where the shorter side is resized to 480 pixels) and the original 4K resolution. To train efficiently, we perform the training at 480p resolution and use the original 4K resolution only for testing. PPR10K includes 11,161 portrait images, each retouched by three experts. We conduct our experiments using the results retouched by Expert A. According to the official setup, we have 8,875 pairs for training and 2,286 pairs for testing. All images are used in a resized format at 360p. We evaluate each method using PSNR, SSIM [34], and the L2-distance in CIE LAB color space ($\Delta E_{ab}$). When measuring runtime, we use the NVIDIA RTX A6000 GPU.

In the IA-NILUT, we employ an MLP consisting of five fully connected layers. The hidden features within this MLP are 256-dimensional, and we utilize the hyperbolic tangent as our activation function. For the parameter prediction model F, a five-layer CNN is used on FiveK, and ResNet18 [10] is applied to PPR10K, following the configurations reported in previous studies [20, 35].

Inspired by the basic filters in Adobe Lightroom, we define five filter names and employ five corresponding guiding prompt pairs as outlined in Table 1. For the Contrast filter, we use different prompts for each dataset, tailoring them to achieve the desired effects.

### 5.2 Visualization of Filter Effects

To demonstrate that the proposed method achieves interpretable filter effects, we visualize filter effects in Figure 4. In these visualizations, only certain parameters are varied while others are held constant at 0. These results indicate that each filter produces a specific effect associated with the corresponding guiding prompts. Figure 5 shows examples of sequential application of predicted parameters, where the enhancement process is visualized in a way that is easy for humans to understand. The sequential application of the filter effects in Figure 5 is for visualization purposes only, and the all filter effects are applied simultaneously in practice.

### 5.3 Ablation Studies

**Filter architecture.** We use the IA-NILUT for a highly expressive filter architecture. To assess the significance of the IA-NILUT, we train 3D LUTs [37] instead of the IA-NILUT using the prompt

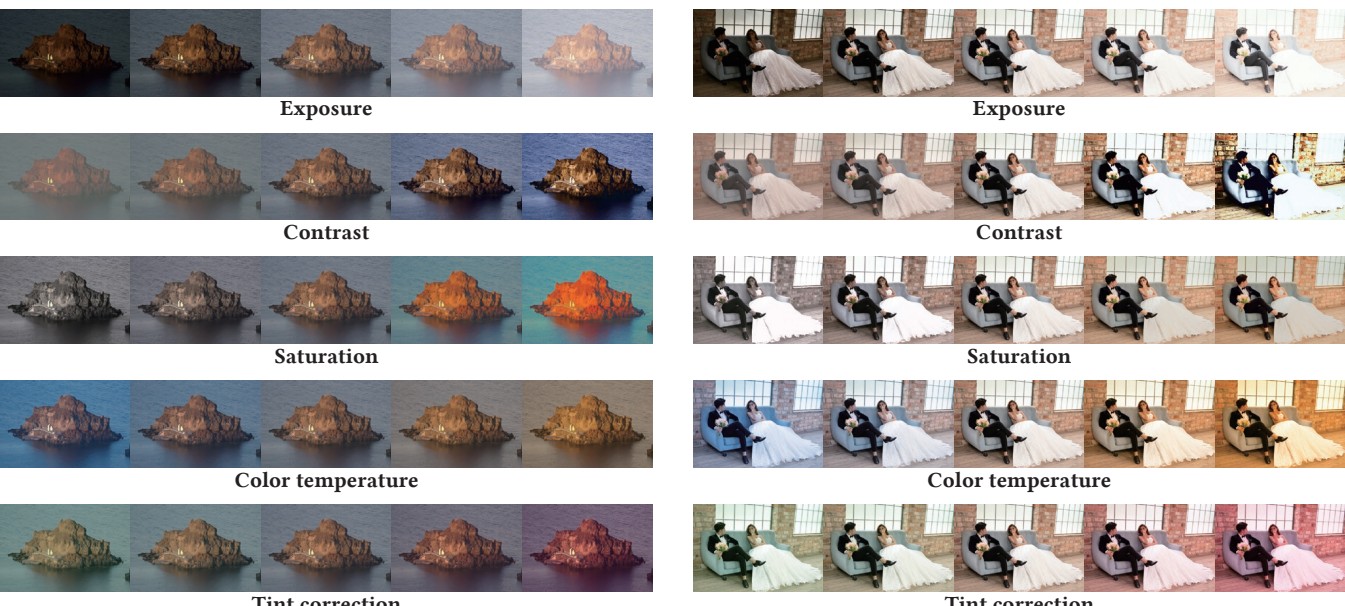

**Figure 4: Visualization of learned filter effects. Only certain parameters are varied while others are held constant at 0. The images on the left and right are samples from FiveK and PPR10K, respectively.**

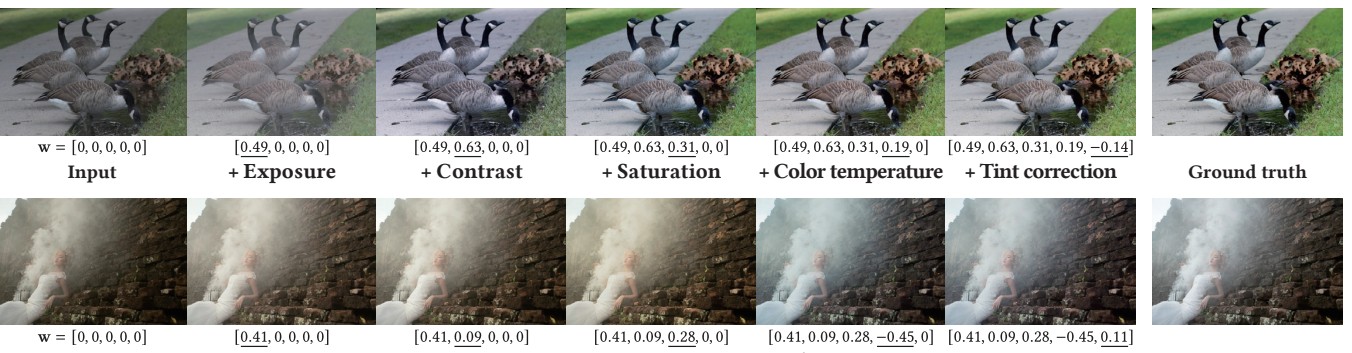

**Figure 5: Sequential application of predicted parameters. This sequential application is for visualization purposes only, and the all effects are applied simultaneously in practice. The top and bottom images are samples from FiveK and PPR10K, respectively.**

guidance loss. To ensure the original image is preserved when **w** is set to **0**, we modify Eq. (1) as follows,

$$\mathbf{Y} = \mathbf{X} + \text{Lookup}(\mathbf{X}, \{\mathbf{I}, \mathbf{O}_1\}) \times w_1 + \cdots$$
$$+ \text{Lookup}(\mathbf{X}, \{\mathbf{I}, \mathbf{O}_J\}) \times w_J. \quad (11)$$

As shown in Table 2, the IA-NILUT achieves higher performance, indicating the higher expressive power of the IA-NILUT. The filter effects of the 3D LUTs trained with the prompt guidance loss are shown in Figure 6. The desired filter effects are not achieved, indicating that the IA-NILUT is essential for interpretable filters.

**LUT bypassing.** We use the LUT bypassing to reduce the computational cost. To demonstrate the effectiveness of the LUT bypassing, we present a comparison of PSNR and runtime in Table 3, and a comparison of the required GPU memory in Figure 7. When processing some 4K images that require more memory than the available limit

in Table 3, we divide the image into four patches and sequentially apply the MLP to each patch. Given that an MLP is computationally intensive, the absence of the LUT bypassing leads to increased computational costs, particularly when processing large-sized images. In contrast, by employing the LUT bypassing, the MLP is only applied to sampling points, the size of which are independent of the overall image size. In addition, the LUT bypassing has little effect on the PSNR. This approach leads to computationally efficient enhancement that is nearly unaffected by the image size.

**Prompt guidance loss.** We employ the prompt guidance loss to assign interpretable names to each filter. To demonstrate the significance of the prompt guidance loss, we present the effects of filters when training the IA-NILUT without this loss in Figure 8. It is difficult to assign interpretable names to these filters. For instance, the first filter influences both exposure and color simultaneously.

**Table 2: Comparison of filter architecture using FiveK (480p).**

| Method | PSNR↑ | SSIM↑ | $\Delta E_{ab}$↓ |
|---|---|---|---|
| 3D LUTs [37] w/ prompt guidance loss | 24.92 | 0.924 | 8.23 |
| IA-NILUT w/ prompt guidance loss | 25.22 | 0.930 | 7.76 |

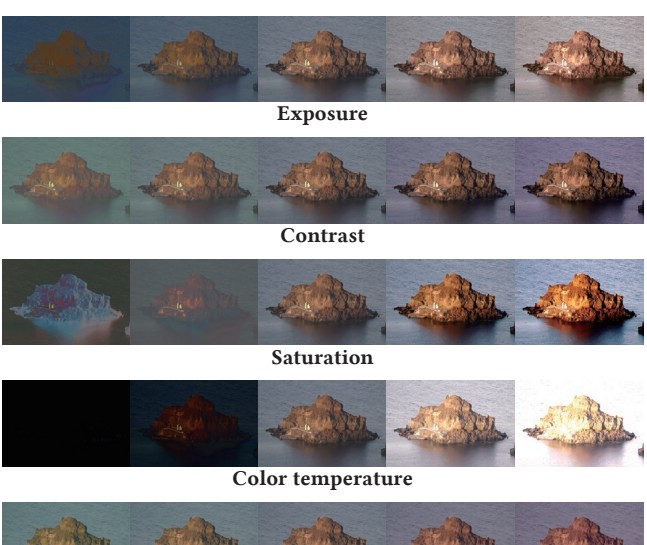

**Exposure**

**Contrast**

**Saturation**

**Color temperature**

**Tint correction**

**Figure 6: Filter effects of 3DLUTs [37] trained with the prompt guidance loss.**

**Table 3: Effectiveness of the LUT bypassing on FiveK.**

| Method | 480p | | Full Res. (4K) | |
|---|---|---|---|---|
| | PSNR↑ | Runtime↓ | PSNR↑ | Runtime↓ |
| Ours w/o LUT bypassing | 25.22 | 1.9 ms | 25.06 | 7.8 ms |
| Ours w/ LUT bypassing | 25.22 | 1.9 ms | 25.05 | 2.0 ms |

Similarly, the second filter affects exposure and saturation together, while the fourth filter impacts color and contrast at the same time. Both the third and fifth filters are able to modify the image's contrast; if both filters had the same "Contrast" name, users would be confused. These results highlight the prompt guidance loss's critical role to assign interpretable names to each filter.

## 5.4 Comparison with the State-of-the-Arts

We employ four interpretable methods: D&R [24], Exposure [11], UIE [15], and RSFNet [23]. For a fair comparison, we utilize only the filters adopted in these methods and apply the same parameter predictor as ours. For the filters from UIE, we exclude non-differentiable filters. Additionally, we include three uninterpretable methods: our baseline method (3D LUTs [37]), and the two state-of-the-art methods (AdaInt [35] and CLUTNet [39]). Since the pre-trained weights for CLUTNet with PPR10K are not publicly available, we only show the performance on FiveK. The performance of these

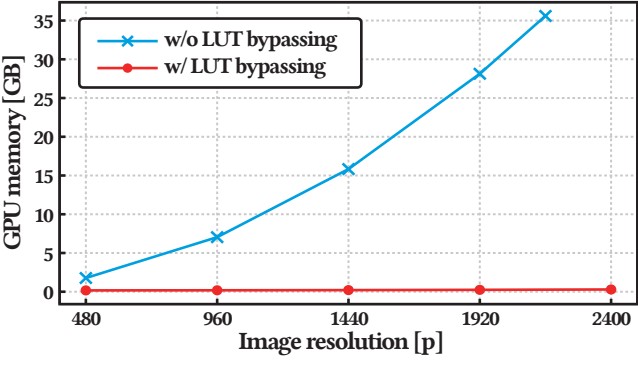

**Figure 7: Required GPU memory w/ and w/o LUT bypassing.**

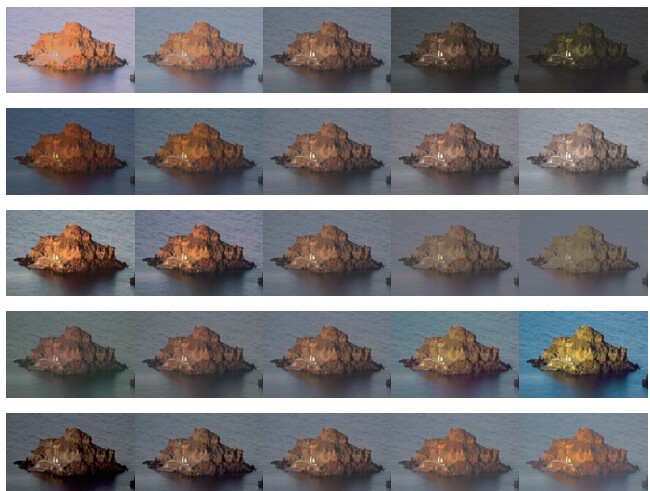

**Figure 8: Filter effects of the IA-NILUT without the prompt guidance loss.**

uninterpretable methods is provided solely for reference, as our primary focus is on interpretable image enhancement.

We present quantitative comparisons in Table 4 and visual comparisons with other interpretable methods in Figure 9. Because our filters are learnable and optimized to predict the ground truth, our method achieves better performance than other predefined filter-based methods. While the runtime for Exposure's filters and UIE's filters is long due to their complex color transformations, the runtime of our method is almost unaffected by the image size thanks to the LUT bypassing. Our method achieves comparable performance to that of uninterpretable methods on some metrics. These results highlight the potential of our method to bridge the gap between interpretability and high performance.

## 5.5 Various Filter Effects

By using different guiding prompts, we can achieve various filter effects. In addition to the guiding prompts listed in Table 1, we assign additional guiding prompts to $w_6$ and then train the filters using only the prompt guidance loss. Figure 10 displays examples of some guiding prompts and their corresponding filter effects. Our

**Table 4: Quantitative comparisons on (a) FiveK and (b) PPR10K. The top three methods are uninterpretable methods, while the bottom five are interpretable methods.**

| Method | (a) FiveK | | | | | | | | (b) PPR10K | | |
|---|---|---|---|---|---|---|---|---|---|---|---|
| | 480p | | | | Full Resolution (4K) | | | | 360p | | |
| | PSNR↑ | SSIM↑ | $\Delta E_{ab}$↓ | Runtime↓ | PSNR↑ | SSIM↑ | $\Delta E_{ab}$↓ | Runtime↓ | PSNR↑ | SSIM↑ | $\Delta E_{ab}$↓ |
| 3DLUTs [37] | 25.36 | 0.927 | 7.56 | 1.5 ms | 25.32 | 0.933 | 7.61 | 1.5 ms | 26.29 | 0.961 | 6.58 |
| AdaInt [35] | 25.50 | 0.930 | 7.47 | 1.5 ms | 25.50 | 0.935 | 7.46 | 1.5 ms | 26.29 | 0.961 | 6.59 |
| CLUTNet [39] | 25.55 | 0.931 | 7.50 | 1.9 ms | 25.50 | 0.935 | 7.53 | 2.1 ms | - | - | - |
| D&R's filters [24] | 23.86 | 0.903 | 9.07 | **1.9 ms** | 23.76 | 0.907 | 9.16 | **1.9 ms** | 24.27 | 0.934 | 8.11 |
| Exposure's filters [11] | 25.04 | 0.920 | 7.83 | 4.3 ms | 24.91 | 0.924 | 7.92 | 15.9 ms | 25.53 | 0.954 | 7.55 |
| UIE's filters [15] | 24.74 | 0.923 | 8.06 | 5.0 ms | 24.61 | 0.928 | 8.14 | 58.9 ms | 25.45 | 0.956 | 7.53 |
| RSFNet's filters [23] | 24.86 | 0.924 | 7.89 | 2.8 ms | 24.82 | 0.928 | 7.96 | 2.8 ms | 25.41 | 0.946 | 7.48 |
| PG-IA-NILUT (ours) | **25.22** | **0.930** | 7.76 | **1.9 ms** | **25.05** | **0.934** | 7.88 | 2.0 ms | **26.00** | 0.957 | 6.81 |

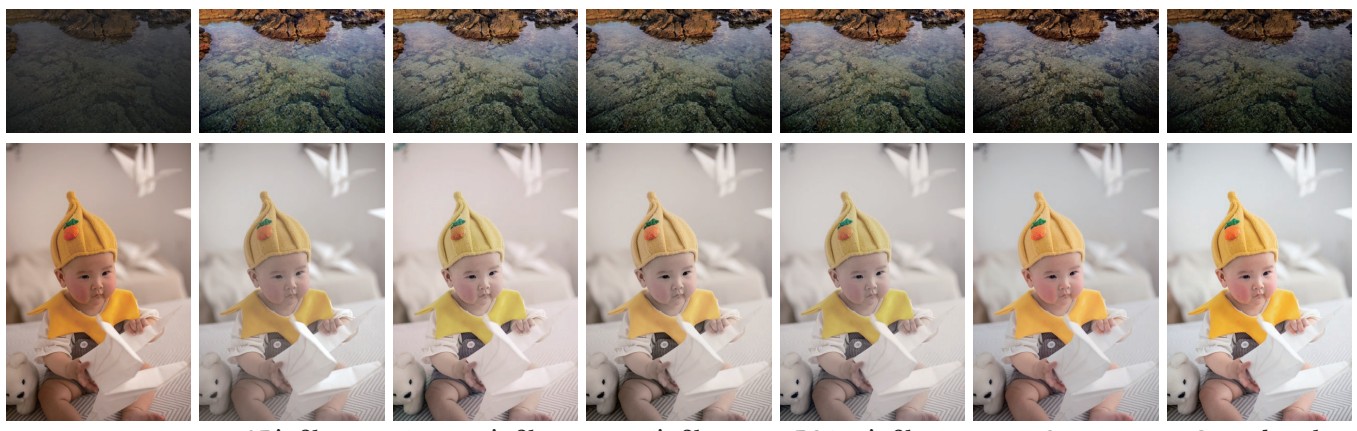

| Input | D&R's filters | Exposure's filters | UIE's filters | RSFNet's filters | Ours | Ground truth |
|---|---|---|---|---|---|---|

**Figure 9: Visual comparisons of interpretive methods, with the top image from FiveK and the bottom from PPR10K.**

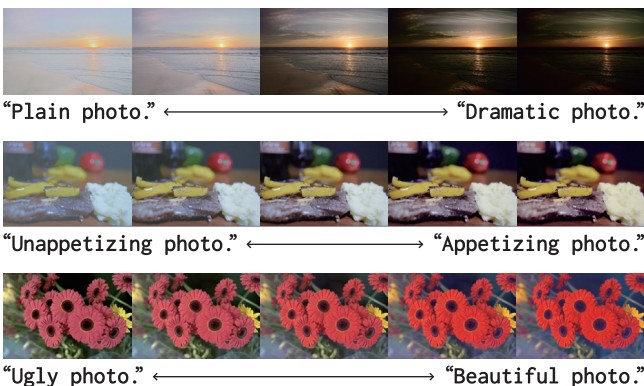

"Plain photo." ←——————————————→ "Dramatic photo."

"Unappetizing photo." ←——————→ "Appetizing photo."

"Ugly photo." ←—————————————→ "Beautiful photo."

**Figure 10: Filter effects by various guiding prompts.**

**Table 5: Impact of the prompt guidance loss on FiveK.**

| Method | PSNR↑ | SSIM↑ | $\Delta E_{ab}$↓ |
|---|---|---|---|
| Ours w/o prompt guidance loss | 25.46 | 0.930 | 7.60 |
| Ours w/ prompt guidance loss | 25.22 | 0.930 | 7.76 |

guidance loss slightly deteriorates performance as shown in Table 5. A potential approach to improve performance while preserving interpretability involves refining the selection of the guiding prompts. We selected the prompts listed in Table 1 heuristically; however, it remains uncertain whether they are optimal for both interpretability and performance. The development of an automatic prompt selection mechanism is identified as an avenue for future research.

## 7 CONCLUSUION

In this paper, we explored interpretable image enhancement. We proposed a highly expressive filter architecture named an IA-NILUT. Additionally, we introduced the prompt guidance loss to assign interpretable names to each filter. Our experiments demonstrated that our method not only provides interpretability but also achieves higher performance compared to existing interpretable filters.

filter is highly expressive, enabling us to achieve various effects and demonstrating its practical utility for image editing applications.

## 6 LIMITATION

Although our method achieves interpretable and high-performing enhancement, it encounters a drawback where the use of the prompt

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
