# OpenReview forum: "Prompt-Guided Image-Adaptive Neural Implicit Lookup Tables for Interpretable Image Enhancement"
_acmmm.org/ACMMM/2024/Conference — MM2024 Poster_

### Official Review · Reviewer_xJHh · 2024-05-18

**Rating:** 4
**Confidence:** 3

**Summary:**

The paper is about a method called Prompt-Guided Image-Adaptive Neural Implicit Lookup Tables (PG-IA-NILUT) for interpretable image enhancement. The main focus of the paper is to propose a highly expressive filter architecture called IA-NILUT, which utilizes implicit neural representations for color transformation. The paper also introduces a prompt guidance loss to assign interpretable names to each filter. The objective of the paper is to achieve high-performing and interpretable enhancement methods for image processing.

**Strengths:**

(1) The paper proposes a filter architecture called IA-NILUT, which incorporates image-adaptive parameters directly into the input features. This approach allows for highly expressive filter effects by capturing nonlinear and intricate relationships between input and output variables using an MLP.
(2) The paper compares its IA-NILUT method with existing LUT-based methods, such as AdaInt and CLUTNet, and highlights the significant difference in the number of image-adaptive parameters. While AdaInt and CLUTNet use 99 and 20 image-adaptive parameters respectively, IA-NILUT achieves similar expressiveness with only five image-adaptive parameters. This makes the IA-NILUT architecture easier to understand and interpret.
(3) The paper also introduces the concept of LUT bypassing to address the computational cost associated with MLPs. By converting the MLP into an LUT and applying it to a smaller image representation, the computational cost remains constant even when processing large-sized images.

**Limitations:**

(1) The lack of accessible code raises concerns about the technical feasibility and replicability of the study.
(2) I am not sure whether the performances shown in Figure 2(b) and (c) are equivalent, apart from their running time.
(3) The utilization of Prompt Guidance Loss appears to be a prevalent strategy, while the level of innovation seems inadequate.

**Suitability:**

3

---

### Official Review · Reviewer_mibs · 2024-05-25

**Rating:** 4
**Confidence:** 2

**Summary:**

This work highlights the limitations of 3D LUTs: (1) the expressive power is limited because multiple 3D LUTs are merely summed in a linear fashion; (2) 3D LUTs lack interpretable names. Since they are optimized solely for predicting target enhancement results, their effects may not be intuitively understood by humans. To address these limitations, this work proposes learnable and interpretable filters named Prompt-Guided Image-Adaptive Neural Implicit Lookup Tables. Specifically, it introduces a novel learnable filter architecture called IA-NILUT. Additionally, it proposes a prompt guidance loss to assign interpretable names to each filter.

**Strengths:**

1. The idea of employing the pretrained CLIP to make the filter interpretable is interesting.

2. The proposed method outperforms its interpretable competitors.

**Limitations:**

1. This work points out that merely adjusting the enhancement effect in a linear manner is one of the main limitations of 3D LUTs. However, as shown in Table 4, 3D LUTs outperform the proposed method over both datasets in terms of PSNR. It is essential to clarify this issue.

2. Interpretability is mainly introduced by the prompt guidance loss. Can this strategy be applied to make other methods interpretable, such as integrating prompt guidance into 3D LUTs?

3. In Section 4.2, why can the proposed method update a specific score while keeping other scores unchanged? It is crucial to provide a visual comparison to demonstrate the effectiveness of the proposed solution. For example, visualize the effect of each filter name separately in Fig.5.

4. It is suggested that the discussion of interpretable and uninterpretable methods be included in the related work. Why is interpretability important, and is it worth compromising performance to make a method interpretable?

**Suitability:**

2

---

### Official Review · Reviewer_eUMQ · 2024-06-02

**Rating:** 3
**Confidence:** 2

**Summary:**

The paper addresses the problem of enhancing image quality with interpretable and customizable filters, which is important for user-friendly image editing. The authors propose an innovative filter architecture using an image-adaptive neural implicit lookup table and a prompt guidance loss to assign understandable names to filters. Their method outperforms existing predefined filter-based approaches, as demonstrated by experimental results showing superior target prediction.

**Strengths:**

The problem of interpretable and learnable filters is important for user-friendly and customizable image editing.
The idea of developing the IA-NILUT, a highly expressive filter architecture, is innovative and well-conceived.
The proposed method introduces a prompt guidance loss to assign interpretable names to each filter, enhancing interpretability.
The results show promising improvements in appearance evaluation metrics like PSNR and SSIM, and also demonstrate reduced runtime compared to chosen baselines.

**Limitations:**

In line 227, Y = Lookup(X, {I, O}), if X is of size H x Y x 3 and I, O are of size H/l x Y/l x 3, how is the corresponding index for a pixel in I determined? Is it just by rounding to the nearest index?
In line 317, it is mentioned that image-adaptive parameters directly into the input features. What do you refer to by image features here? Are they r^{x}, g^{x}, b^{x}? This is not clear, and Eq. 3 does not explain this.
An ablation study on employing various values of the hyperparameter N is missing. In line 376, it is stated, In our experiment, we set 𝑁 to 33, resulting in I being treated as an image composed of 35,937 pixels.
As mentioned in Fig 1, the input image is downsampled. However, it is unclear from the text and equations how the downsampled input image leads to an output image that matches the input size. While implicit representation is employed, it is not clear where and how it is used. Please clarify.
In line 393, for example, AdaInt and CLUTNet use 99 and 20 image-adaptive parameters, respectively. However, this approach makes interpretability more complex. Too many parameters can make the image editing process confusing for users. In contrast, our method boosts expressiveness by using an implicit neural representation, without increasing the number of image-adaptive parameters. Please specify how many image-adaptive parameters the proposed approach employs.
It is not clear how the MLP weights and five filter names are related. The method mentions employing five filters as indicated in Table 1.
Why was [7] not used as a baseline? Since this work builds on it, it is important to compare with it to validate the contributions. Given that their code is publicly available, it is unclear why this important work was omitted in the comparisons.

Also, how does the work relate to 'Iterative Prompt Learning for Unsupervised Backlit Image Enhancement (ICCV'23)'

**Suitability:**

3

---

### Meta-Review · Area_Chair_3G7Q · 2024-07-06

**Recommendation:** Accept (Poster)
**Confidence:** 4

**Metareview:**

The authors  propose the  Prompt-Guided Image-Adaptive Neural Implicit Lookup Tables (PG-IA-NILUT) for interpretable image enhancement through an expressive filter architecture called IA-NILUT along with a prompt guidance loss to assign interpretable names to each filter.

The reviewers agree on the suitability of the paper to the ACM MM conference along with several strengths (filter architecture, interpretability, experimental results, etc). The reviewers also identify several limitations in the submission (some editorial and clarity issues, some missing related work, missing visual comparison, etc.)  It is recommended that this paper be accepted for a paper presentation assuming the editorial issues will be fixed in the camera ready version.